# New Species of Large-Spored *Alternaria* in Section *Porri* Associated with Compositae Plants in China

**DOI:** 10.3390/jof8060607

**Published:** 2022-06-06

**Authors:** Lin Zhao, Huan Luo, Hong Cheng, Ya-Nan Gou, Zhi-He Yu, Jian-Xin Deng

**Affiliations:** 1Department of Plant Protection, College of Agriculture, Yangtze University, Jingzhou 434025, China; zhaolin999@hotmail.com (L.Z.); hongcheng7777@hotmail.com (H.C.); gynan024@hotmail.com (Y.-N.G.); 2Department of Applied Biology, Chungnam National University, Daejeon 34134, Korea; luohuan_0813@163.com; 3Department of Biology, College of Life Sciences, Yangtze University, Jingzhou 434025, China; zhiheyu@hotmail.com

**Keywords:** *Alternaria*, compositae, morphology, multi-locus sequence analyses, taxonomy

## Abstract

*Alternaria* is a ubiquitous fungal genus including saprobic, endophytic, and pathogenic species associated with a wide variety of substrates. It has been separated into 29 sections and seven monotypic lineages based on molecular and morphological data. *Alternaria* sect. *Porri* is the largest section, containing the majority of large-spored *Alternaria* species, most of which are important plant pathogens. Since 2015, of the investigations for large-spored *Alternaria* species in China, 13 species were found associated with Compositae plants based on morphological comparisons and phylogenetic analyses. There were eight known species and five new species (*A. anhuiensis* sp. nov., *A. coreopsidis* sp. nov., *A. nanningensis* sp. nov., *A. neimengguensis* sp. nov., and *A. sulphureus* sp. nov.) distributed in the four sections of *Helianthiinficientes*, *Porri*, *Sonchi*, and *Teretispora*, and one monotypic lineage (*A. argyranthemi*). The multi-locus sequence analyses encompassing the internal transcribed spacer region of rDNA (ITS), glyceraldehydes-3-phosphate dehydrogenase (GAPDH), *Alternaria* major allergen gene (Alt a 1), translation elongation factor 1-alpha (TEF1), and RNA polymerase second largest subunit (RPB2), revealed that the new species fell into sect. *Porri*. Morphologically, the new species were illustrated and compared with other relevant large-spored *Alternaria* species in the study. Furthermore, *A. calendulae*, *A. leucanthemi*, and *A. tagetica* were firstly detected in *Brachyactis ciliate*, *Carthamus tinctorius*, and *Calendula officinalis* in China, respectively.

## 1. Introduction

*Alternaria* is a cosmopolitan and widely distributed fungal genus described originally by Nees (1816), which is characterized by the dark-coloured phaeodictyospores in chains and a beak of tapering apical cells [1]. It is also associated with nearly every environmental substrate including animal, plant, agricultural product, soil, and the atmosphere. Species of *Alternaria* are known as serious plant pathogens, causing enormous losses on many crops [1,2]. The taxonomy is mainly based on sporulation patterns and their conidial shape, size, and septation [2,3]. Around 280 species are summarised and recognised on the basis of morphology [2], comprising two groups, large-spored (60–100 μm long conidial body) and small-spored (below 60 μm conidial body) [4,5,6].

Since the 20th century, molecular approaches, especially multi-locus phylogenetic analyses, have been used to identify *Alternaria* species [7,8,9,10]. Over ten gene regions are used in the classification, such as the internal transcribed spacer region of rDNA (ITS), large subunit ribosomal DNA (LSU), mitochondrial small subunit (mtSSU), glyceraldehydes-3-phosphate dehydrogenase (GAPDH), *Alternaria* major allergen gene (Alt a 1), translation elongation factor 1-alpha (TEF1), RNA polymerase second largest subunit (RPB2), and plasma membrane ATPase [1,4,7,9,11,12,13,14,15,16,17,18]. *Alternaria* has been separated into 29 sections and seven monotypic lineages [19,20,21]. The introduction of a molecular phylogenetic approach has helped to clarify their taxonomy, combining many allied genera into one large genus of *Alternaria* complex [1].

Due to the effects of *Alternaria* on humans and their surroundings, the identification is particularly important to agriculture, medicine, and science. The Compositae plants serve as food plant, oil seed, seed plant, ornamental, and sources of medicine and insecticide worldwide [22], of which nearly 3000 species almost 240 genera have been found in China [23]. Most *Alternaria* are commonly plant pathogens leading to substantial economic losses caused by Alternaria leaf spots and defoliation [18,24,25,26]. Large-spored *Alternaria* species encompassing 148 species are almost phytopathogenic demonstrated [2].

During the investigation of large-spored *Alternaria* in China, five new species were encountered from diseased leave samples of composite plants. The objectives of this study were to identify them on the basis of the cultural and conidial morphology incorporate with multi-loci phylogeny (ITS, GAPDH, Alt a 1, TEF1, and RPB2). The present multi-locus analysis supplemented with cultural and morphological data forms an example for *Alternaria* species recognition. The five new species described in this study add species diversity to large-spored *Alternaria* and provide theoretical and practical basis for the further identification and disease management.

## 2. Materials and Methods

### 2.1. Sample Collection and Fungal Isolation

Symptomatic samples of composite plants (14) have been randomly collected from different provinces in China since 2015. For fungal isolation, the samples were put into sterile plastic bags and taken to the laboratory. Small leaf segments (2 mm) with disease lesions were placed into petri dishes with moist filter papers and incubated at 25 °C in dark for conidial sporulation. Single spore of large-spored *Alternaria* was picked by a sterilized glass needle under the stereoscopic microscope and transferred to potato dextrose agar (PDA: Difco, Montreal, Canada). Over ten similar spores were randomly picked from a sample for sub-culturing to obtain the pure cultures, and two to three strains were selected for deposition when exhibiting similar cultural morphology on PDA. A total of 81 strains were kept in test-tube slants and deposited at 4 °C. Living ex-type strains were preserved in the Fungi Herbarium of Yangtze University (YZU), in Jingzhou, Hubei, China.

### 2.2. Morphological Observations

To determine cultural characteristics including growth rate, color and texture of colonies [27], mycelial plugs (6 mm in diameter) were taken from the edge of colonies grown on PDA. Then, the plugs were put on fresh PDA plates (90 mm) at 25 °C for 7 days in darkness. To observe the conidial morphology (conidial sporulation patterns, shape, size, etc.), mycelia were grown on potato carrot agar (PCA) and V8 juice agar (V8A) inoculated at 22 °C with a light period of 8 h light/16 h dark [2]. After 7 days, conidia and sporulation patterns were observed. Conidiophores and conidia were mounted with lactophenol picric acid solution and photographed with a Nikon ECLIPSE Ni-U microscope (Nikon, Japan). Randomly selected conidia (*n* = 50) were separately measured for each characterization.

### 2.3. DNA Extraction and PCR Amplification

Genomic DNA extraction was performed using fresh mycelia collected from colonies grown on PDA [28]. Polymerase chain reaction (PCR) amplifications of the internal transcribed spacer region of rDNA (ITS), glyceraldehydes-3-phosphate dehydrogenase (GAPDH), *Alternaria* major allergen gene (Alt a 1), translation elongation factor 1-alpha (TEF1), and RNA polymerase second largest subunit (RPB2) gene regions were amplified with the primer pairs ITS5/ITS4 [29], EF1-728F/EF1-986R [30], gpd1/gpd2 [31], Alt-for/Alt-rev [12], and RPB2-5F2/RPB2-7cR [32,33], respectively. A 25 μL of the PCR reaction mixture comprising 21 μL of 1.1 × Taq PCR Star Mix (TSINGKE, Beijing, China), 2 μL template DNA and 1 μL of each primer was applied and performed in a BIORAD T100 thermocycler [1]. Successfully amplified PCR products were purified and sequenced by TSINGKE company (Beijing, China).

### 2.4. Phylogenetic Analyses

The resulted sequences were examined by BioEdit v.7.0.9 [34] and assembled with PHYDIT 3.2 [35]. All newly generated sequences were deposited in GenBank (Table 1). Relevant sequences [4] were retrieved from NCBI database based on the results of BLAST searches (Table 1). The concatenated sequence dataset of multiple loci was aligned using MEGA v.6.0 [36]. Phylogenetic analyses of each alignment were performed using maximum likelihood (ML) and Bayesian inference (BI) methods. ML analysis was conducted using RAxML v.7.2.8 [37]. Bootstrapping with 1000 replicates was performed using the model of nucleotide substitution obtained by MrModeltest. For the BI analysis, it was performed using parameters including 1,000,000 Markov chain Monte Carlo (MCMC) algorithm with Bayesian posterior probabilities [38]. MrModel test v.2.3 used the best-fit model (GTR+I+G) according to the Akaike Information Criterion (AIC). Two MCMC chains were run from random trees for 10^6^ generations, and the trees were sampled every 100th generation. After discarding the first 25% of the samples, the 50% majority rule consensus tree and posterior probability values were calculated. Finally, the resulting trees were edited in FigTree v.1.3.1 [39]. Branch support of the groupings (>60%/0.6 for ML bootstrap value-BS/posterior probability-PP) were indicated in the phylogram. *Alternaria gypsophilae* CBS 107.41 in sect. *Gypsophilae* was used as an outgroup.

## 3. Results

In the present study, large-spored *Alternaria* species associated with Compositae leaf spot in China since a survey from 2015 are summarized based on the phylogenetic analysis of GAPDH and RPB2 gene fragments (Appendix A). A total of 13 species including the present five new taxa revealed in four sections of *Helianthiinficientes* (*A. helianthiinficiens*), *Porri* (*A. calendulae*, *A. tagetica* and *A. zinniae*), *Sonchi* (*A. cinerariae* and *A. sonchi*), and *Teretispora* (*A. leucanthemi*), and one monotypic lineage (*A. argyranthemi*) (Appendix A). Meanwhile, a comprehensive description of the five new species in sect. *Porri* are described as *A. anhuiensis* sp. nov., *A. coreopsidis* sp. nov., *A. nanningensis* sp. nov., *A. neimengguensis* sp. nov., and *A. sulphureus* sp. nov..

### 3.1. Phylogenetic Analysis

The multi-gene phylogeny was constructed to determine the accurate positions of the new *Alternaria* based on five sequence loci (ITS + GAPDH + Alt a 1 + TEF1 + RPB2) (Table 1). The analysis comprised sequences of the ITS (504 characters), GAPDH (526 characters), Alt a 1 (457 characters), TEF1 (342 characters), and RPB2 (672 characters) gene regions with a total length of 2501 characters. The tree topologies (Figure 1) computed from the ML and BI analyses, were similar to each other, resulting in identical species-clades and the ML topology was presented as basal tree. The present strains fell into five separate branches in sect. *Porri* of *Alternaria*. Strain YZU 171206 was sister to *A. alternariacida* supported with a PP value of 1.0, which close to *A. silybi* with low BS and PP values surpport. Strains YZU 161159 and YZU 161160 formed an independent clade (BS/PP = 100%/1.0). Strain YZU 171523 fell into an individual branch close to *A. obtecta* and *A. tillandsiae* well-supported by 97%/0.99 (BS/PP). Strain YZU 171784 was clustered with *A. cirsinoxia*, *A. centaureae*, *A. cichorii*, and *A. cantannaches* supported by values of 79%/1.0 (BS/PP). Strain YZU 191448 was out group of strain YZU 171206, *A. silybi* and *A. alternariacida* with BS and PP values below 60% and 0.6. The results indicated that the five branches represent five new species from three different hosts (*Coreopsis basalis*, *Cosmos sulphureus*, and *Lactuca seriola*).

### 3.2. Taxonomy

***Alternaria anhuiensis*** H. Luo and J.X. Deng, sp. nov. (Figure 2).

**MycoBank No**: 844033.

**Etymology****:** Named after the collecting locality, Anhui Province.

**Typification****:** China, Anhui Province, Hefei City, from leaf spot of *Coreopsis basalis*. June, 2017, J.X Deng, ex-type culture YZU 171206.

**Description****:** Colonies on PDA circular, buff in the centre, flocculent with brown halo at the edge; reverse crimson pigment at centers, light yellow at margins, 59–60 mm in diam, at 25 °C for 7 days. On V8A, conidiophores arising from substrate or lateral of aerial hyphae with geniculate conidiogenous loci at or near apex, straight or curved, smooth-walled, septate, pale to dark brown, (40–) 60–145 (–203) × (4.5–) 5–7.5 (–8) μm; conidia solitary, long-narrow ovoid or ellipsoid body, apex rounded, base narrow, smooth-walled, single to double beak, dark brown, 61–100 (–111.5) × (11.5–) 13–19.5 μm, 6–11 transverse septa, 0–1 (–2) longitudinal septa; beak long-narrowed filiform, 1-beak, (32–) 58–133 (–150.5) × 2.5–4 (–4.5) μm; 2-beak, (22–) 60.5–90.5 (–116.5) × 2.5–3.5 μm. On PCA, conidiophores straight or curved, smooth-walled, septate, (42.5–) 50–140 × 4.5–6.5 (–9) μm; conidia solitary, long-narrow ovoid or ellipsoid body, single to double beak, triple or quadruple beaks not common, black brown, (55–) 66–105 × 11–16 μm, 5–10 (–11) transverse septa, 0–1 longitudinal septum; beak long-narrowed filiform, 1-beak, 100–180 (–202) × 2.5–4 μm; 2-beak, 95–217 (–236) × 2.5–4 (–5.5) μm; 3-beak, 60–140 × 2.5–3.5 μm; 4-beak (*n* = 1), 82 × 3 μm.

**Notes****:** Phylogenetic analysis of the species based on a combined dataset of ITS, GAPDH, Alt a 1, TEF1, and RPB2 gene fragments falls in an individual clade close to *A. alternariacida* and *A. silybi* in sect. *Porri* (Figure 1). Morphologically, its primary conidiophores can generate geniculate conidiogenous loci at or near apex which differed from those two species (Figure 2, Table 2). It can be easily distinguished from *A. alternariacida* by producing more transverse septa and shorter beaks. Moreover, its conidia are solitary while *A. alternariacida* forms solitary or in unbranched chains of 2 (–3) conidia.

***Alternaria coreopsidis*** H. Luo and J.X. Deng, sp. nov. (Figure 3).

**MycoBank No**: 844034.

**Etymology:** Named after the host genus name, *Coreopsis*.

**Typification:** China, Shaanxi Province, Xian City, from leaf spot of *Coreopsis basalis*. June, 2016, J.X Deng, ex-type culture YZU 161160.

**Description:** Colonies on PDA circular, buff halo in the centre, villiform with white at the edge; reverse dark brown at centers, vinaceous buff pigment at margins, 47–48 mm in diam, at 25 °C for 7 days. On V8A, conidiophores arising from substrate or lateral of aerial hyphae, solitary, simple, straight to slightly curved, septate, pale to dark brown, apical conidiogenous locus, pale brown, (34–) 50–86 (–115.5) × 5–7 (–9) μm; conidia solitary or in unbranched chains of 2 conidia, long-narrow ovoid or ellipsoid body, smooth-walled, single beak, yellow or brown, (48.5–) 55–80 (–85) × (9–) 10–15 μm, 6–9 transverse septa, 0–1 longitudinal septa; beak filamentous, 1-beak, (20–) 30–140 (–206) × (2–) 2.5–4 μm; normally, false beak swollen at the apex, around 8–10.5 (–14) × 4.5–5 (–6) μm. On PCA, conidiophores straight or curved, smooth-walled, septate, (24–) 50–90 (–135) × 5–7.5 (–9) μm; conidia long-narrow ovoid or ellipsoid body, apex rounded, single beak, pale brown, (40–) 45–70 × 9–13 μm, (5–) 6–8 (–9) transverse septa, 0–1 longitudinal septa; beak filamentous, 1-beak, (0–) 15–100 (–175) × (0–) 2–4 μm; swollen apex of false beak commonly 10–13 (–16.5) × 5–6 (–6.5) μm.

**Materials examined:** China, Shaanxi Province, Xian City, from leaf spot of *Coreopsis basalis*. June 2016, J.X Deng, living culture YZU 161159.

**Notes:** Phylogenetically, the species falls into an independent lineage outside of a clade comprising type species of *A. porri* of sect. *Porri* (Figure 1). It can be delimited based on either of GAPDH and RPB2 gene sequences (Appendix A). The species is characterized by producing conidia with false beak swollen at the apex up to 8–13 (–16.5) × 4.5–6.5 μm (Figure 3; Table 2).

***Alternaria nanningensis*** H. Luo and J.X. Deng, sp. nov. (Figure 4).

**MycoBank No**: 844035.

**Etymology:** Named after the collecting locality, Nanning City.

**Typification:** China, Guangxi Province, Nanning City, from leaf spot of *Cosmos sulphureus*. July 2017, J.X Deng, ex-type culture YZU 171523.

**Description:** Colonies on PDA irregular, pistac, entire; reverse dark olive green, slightly protuberant with white at margins, 56–57 mm in diam, at 25 °C for 7 days. On V8A, conidiophores arising from substrate or lateral of aerial hyphae with geniculate conidiogenous loci at apex, straight or curved, smooth-walled, septate, pale brown, 38–59 (–64)× 4–5 (–6) μm; conidia solitary, ovoid or ellipsoid body, base narrow, smooth-walled, single beak, pale to yellow brown, (40.5–) 47–79 (–87) × 9–13.5 (–15) μm, 6–10 (–11) transverse septa, 0–1 longitudinal septa; beak long-narrowed filiform, 1-beak, 10–30 × (1–) 1.5–2 (–3) μm. On PCA, conidiophores straight or curved, smooth-walled, septate; 32–70 (–86) × 4–5.5 μm; conidia solitary, ovoid or ellipsoid body, single beak, pale to yellow brown, (49–) 55–77 (–82) × 10.5–13.5 (–15) μm, (5–) 6–9 (–10) transverse septa, 0–1 longitudinal septum; beak long-narrowed filiform, 1-beak, 13–26 (–44) × 1.5–2 (–2.5) μm.

**Notes:** The species is phylogenetically recognized as a distinct species in sect. *Porri* based on ITS, GAPDH, Alt a 1, TEF1, and RPB2 which displays a close relationship with *A. obtecta*, *A. tillandsiae*, and *A. steviae* (Figure 1). Compared with them, it is quite different by producing smaller conidia with short beaks (Figure 4; Table 2). Furthermore, its conidia are smooth-walled while some conidia of *A. obtecta* and *A. steviae* are minutely punctulate. *Alternaria nanningensis* forms simple conidiophores (solitary). But many conidiophores of *A. steviae* produce geniculate extensions and additional conidia, yielding tiny distal clumps of sporulation.

***Alternaria neimengguensis*** H. Luo and J.X. Deng, sp. nov. (Figure 5).

**MycoBank No**: 844036.

**Etymology:** Named after the collecting locality, Inner Mongolia Autonomous Region.

**Typification:** China, Inner Mongolia Autonomous Region, Inner Mongolia Agricultural University, IMAU, from leaf spot of *Lactuca seriola*. September 2017, J.X Deng, ex-type culture YZU 171784.

**Description:** Colonies on PDA circular, pale brown en masse, flocculent, reverse dark olive green at centers, pale brown at margins, 51–54 mm in diam, at 25 °C for 7 days. On V8A, conidiophores arising from substrate or lateral of aerial hyphae, straight or curved, smooth-walled, septate, brown, 26–45 (–51) × 5–7 (–8) μm; conidia solitary, ovoid or ellipsoid body, apex rounded, base wide, smooth-walled, single to double beak, brown, (70–) 77–130 (–143.5) × (13–) 15–20 (–23) μm, 6–11 (–12) transverse septa, 0–1 (–2) longitudinal septa; beak long-narrowed filiform, 1-beak, (24.5–) 35–65 (–76) × (1.5–) 2–3 (–4) μm; 2-beak, (33–) 45–65 (–92) × (2–) 2.5–3 (–3.5) μm. On PCA, conidiophores straight or curved, smooth-walled, septate; 35–70 (–75)× 5–6.5 (–7.5) μm; conidia solitary, ovoid or ellipsoid body, apex rounded, single to double beak, pale to yellow brown, (59–) 66–104 (–120.5) × 13–18 (–20) μm, (5–) 6–10 (–11) transverse septa, 0–1 (–2) longitudinal septa; beak long-narrowed filiform, 1-beak, (13–) 31.5–60 (–93) × 1.5–3 μm; 2-beak, (12–) 26–53 (–80) × 1.5–2.5 (–3) μm.

**Notes:** In the phylogeny, the species is sister to *A. cirsinoxia*, *A. centaureae*, *A. cichorii*, and *A. catananches* (Figure 1). The conidiophores are distinct to *A. cirsinoxia* whose are 2–3 arm branches near a conidiophore tip and progressively geniculate, yielding tufts of several conidia. They are different from *A. cichorii* whose are frequently branch or proliferate in a geniculate manner near the apex, yielding terminal clumps of 4–5 conidia. In conidial morphology, it is obviously different from those four species by producing larger conidia (Table 2).

***Alternaria sulphureus*** L. Zhao and J.X. Deng, sp. nov. (Figure 6).

**MycoBank No:** 844037.

**Etymology:** Named after the host species name, *Cosmos sulphureus*.

**Typification:** China, Shanxi Province, from leaf spot of *Cosmos sulphureus*. September 2019, J.X Deng, ex-type culture YZU 191448.

**Description:** Colonies on PDA circular, light brown in the centre, buff texture velutinous at the edge, reverse black brown at centers, 62–63 mm in diam, at 25 °C for 7 days. On V8A, conidiophores arising from substrate, solitary, simple, straight to slightly curved, septate, apical conidiogenous locus, pale brown; (50–) 63–100 (–108) × 6–8 (–9) μm; conidia solitary, sometimes in chains of two conidia, ovoid, ellipsoid or obovoid body, smooth-walled, pale to yellow, (64–) 74–116 × (12.5–) 14–20 (–25.5) μm, (5–) 7–11 transverse septa, 0–1 (–2) longitudinal septa; beak long-narrowed filiform, 1-beak, (25.5–) 34–151 (–159.5) × 2.5–4.5 (–5.5) μm; 2-beak (*n* = 1), 129 × 4 μm. On PCA, conidiophores straight or curved, smooth-walled, septate, (34.5–) 40.5–56 (–85) × 5–7.5 μm; conidia ovoid, ellipsoid, or obovoid body, apex rounded, single to double beak, triple beaks not common, pale brown, 80–110 × 16–24 μm, 6–10 transverse septa, 0–1 longitudinal septum; beak long-narrowed filiform, 1-beak, (73–) 110–195 × 3–5 μm; 2-beak, (74–) 96–170 × 3–4 μm; 3-beak (*n* = 1), 109.5 × 3.5 μm.

**Notes:** This species is phylogenetically related to *A. silybi*, *A. alternariacida* and *A. anhuiensis* sp. nov. in sect. *Porri* (Figure 1). It could be distinguished from *A. silybi* and *A. alternariacida* by forming larger conidia (Figure 6; Table 2) and is quite different from *A. alternariacida* by producing multiple and shorter beaks.

## 4. Discussion

Thirteen large-spored *Alternaria* species associated with Compositae leaf spot in China were assigned to four sections and one monotypic lineage in this study. Among theses species, five new species (*A. anhuiensis* sp. nov., *A. coreopsidis* sp. nov., *A. nanningensis* sp. nov., *A. neimengguensis* sp. nov., and *A. sulphureus* sp. nov.) were clearly recognized in section *Porri*. The section is speciose assessing encompassing 117 large-spored *Alternaria* [5]. In 2014, the section is reduced 82 morphospecies in to 63 phylogenetic species [2]. They are commonly pathogenic and could induce typical black necrotic lesions surrounded by chlorotic areas. There are some important famous plant pathogens, such as *A. porri* on *Allium* plants (Liliaceae), *A. solani* for potato (Solanaceae), *A. sesami* for sesame (Pedaliaceae) and *A. dauci* for carrot (Umbelliferae) [2]. Twenty-one species are comprised in sect. *Porri* associated with the Compositae family [4]. This study provides new data supplements for the *Alternaria* taxonomy of sect. *Porri*.

Morphologically, large-spored *Alternaria* species in sect. *Porri* are characterised by broadly ovoid, obclavate, ellipsoid, subcylindrical or obovoid, medium to large conidia containing multiple transverse and longitudinal septa, solitary or in short chains with a simple or branched, long to filamentous beak [4]. Among these characteristics, sporulation patterns, conidial body, transverse septa, and beak type provide useful information for the preliminary separation into sections [2]. Morphology is quite important for new fungal species identification, which can be defined based on unique morphological characters when the molecular data is not well-supported [41]. Morphological comparisons of the present new species and their relevant species in sect. *Porri* were conducted (Table 2). For the sporulation patterns, the conidia of *A. anhuiensis*, *A. nanningensis*, *A. neimengguensis*, and *A. sulphureus* are solitary produced except *A. coreopsidis*, which similar to *A. alternariacida*, *A. cichorii*, *A. cirsinoxia*, and *A. steviae* forming chain of 2 (–3) units [2,4]. In conidial morphology, *A. anhuiensis*, *A. nanningensis*, *A. neimengguensis*, and *A. sulphureus* are distinguishable from their closely related species based on the size of conidial bodies (Table 2) and also the wall ornamentations [2,4]. On the other hand, *A. anhuiensis*, *A. neimengguensis*, and *A. sulphureus* are readily be distinguished by producing multiple beaks. By the way, there are no significant differences on conidial morphology of PCA and V8A medium for all species.

In addition, morphological variation and fundamental pleomorphism complicate the *Alternaria* species recognition, and host plants reflect some evidences for the identification [3]. With the discovery of *Alternaria* species, it has been found from various plants of Compositae [1,4,21,42,43]. *Alternaria calendulae* has been reported from *Calendula officinalis* in Czech Republic [2], Germany [4], Japan [4], and Korea [44]. It also is found on *C. officinalis* in China and firstly on *Brachyactis ciliata* in the study. *Alternaria leucanthemi* has previously been found on *Chrysanthemum maximum* from Netherlands [1] and *Helianthus annuus* from China [45]. It is firstly isolated from *Carthamus tinctorius* in this study. In addition, *A. tagetica* is commonly associated with *Tagetes* plants (*Tagetes erecta* and *Tagetes patula*) [3,4,46,47,48], which firstly encountered from *Calendula officinalis* in this study. Interestedly, the five new species are isolated from three different composite hosts (*Coreopsis basalis*, *Cosmos sulphureus*, and *Lactuca seriola*) and *A. cinerariae* are found on five different composite plants in China (Appendix A). The results suggest that an *Alternaria* species may associated with several host plants.

## 5. Conclusions

The present data indeed revealed a diversity of large-spored *Alternaria* associated with Compositae plants in China. A total of 13 large-spored *Alternaria* species were obtained and circumscribed as eight known species and five new species belonging to the four sections of *Helianthiinficientes*, *Porri*, *Sonchi*, and *Teretispora*, and one monotypic lineage (*A. argyranthemi*) based on the morphological characteristics and molecular properties of multiple DNA sequences (ITS, GAPDH, Alt a 1, TEF1, and RPB2). *Alternaria*
*calendulae*, *A. leucanthemi*, and *A. tagetica* were firstly isolated from *Brachyactis ciliate*, *Carthamus tinctorius*, and *Calendula officinalis* in China, respectively. Since large-spored *Alternaria* species are almost demonstrated phytopathogens, further study on the pathogenicity is needed to verify in the future.

## Figures and Tables

**Figure 1 jof-08-00607-f001:**
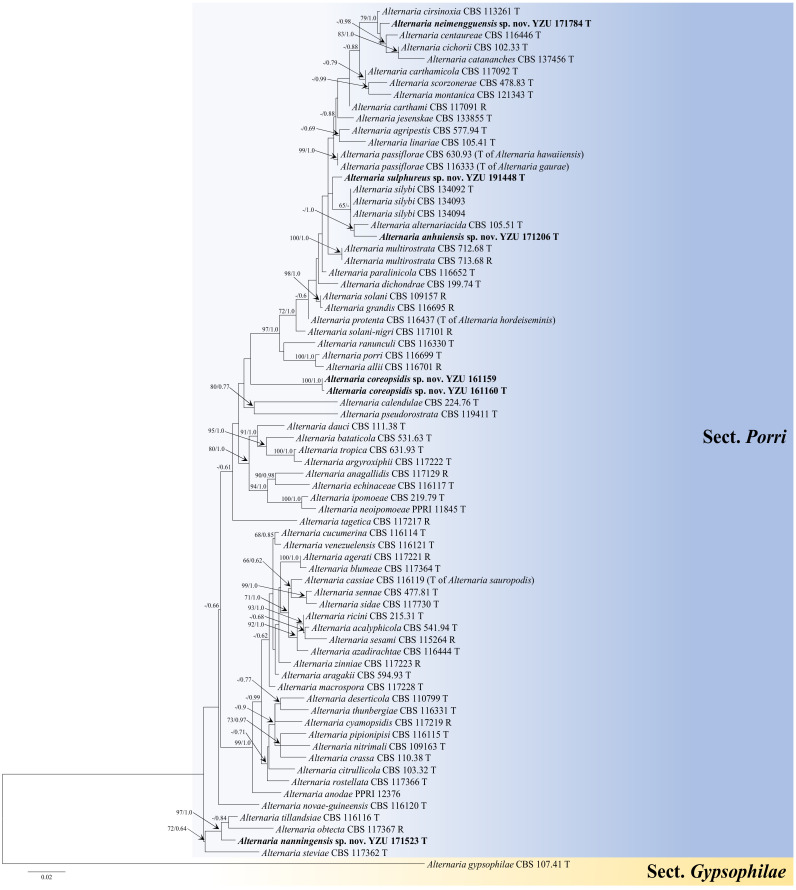
Maximum likelihood (ML) phylogram of new five *Alternaria* species from the Compositae family based on a combined dataset of ITS, GAPDH, Alt a 1, TEF1, and RPB2 gene sequences. The RAxML bootstrap support values >60%(ML) and Bayesian posterior probabilities >0.6 (PP) are given at the nodes (ML/PP). The present strains are in bold.

**Figure 2 jof-08-00607-f002:**
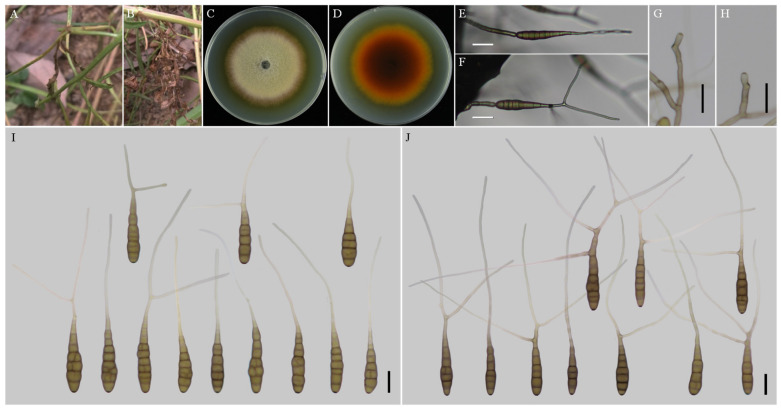
Morphology of *Alternaria anhuiensis* sp. nov. (**A**,**B**) Natural symptoms of *Coreopsis basalis*; (**C**,**D**) Colony phenotypes (on PDA for 7 days at 25 °C); (**E**,**F**) Sporulation patterns (on V8A at 22 °C); (**G**,**H**) Conidiophores (on V8A at 22 °C); (**I**) Conidia (on V8A at 22 °C); (**J**) Conidia (on PCA at 22 °C). Bars: (**E**–**J**) = 25 μm.

**Figure 3 jof-08-00607-f003:**
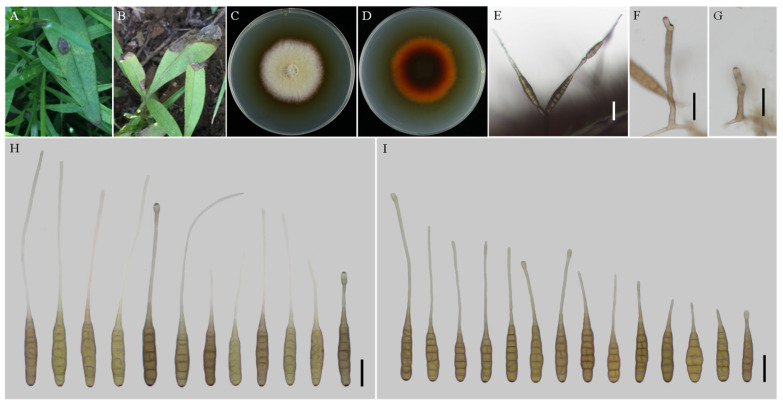
Morphology of *Alternaria coreopsidis* sp. nov. (**A**,**B**) Natural symptoms of *Coreopsis basalis*; (**C**,**D**) Colony phenotypes (on PDA for 7 days at 25 °C); (**E**) Sporulation patterns (on V8A at 22 °C); (**F**,**G**) Conidiophores (on V8A at 22 °C); (**H**) Conidia (on V8A at 22 °C); (**I**) Conidia (on PCA at 22 °C). Bars: (**E**–**I**) = 25 μm.

**Figure 4 jof-08-00607-f004:**
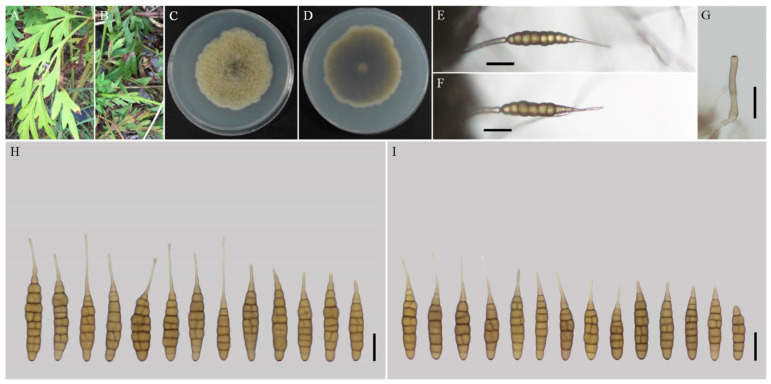
Morphology of *Alternaria nanningensis* sp. nov. (**A**,**B**) Natural symptoms of *Cosmos sulphureus*; (**C**,**D**) Colony phenotypes (on PDA for 7 days at 25 °C); (**E**,**F**) Sporulation patterns (on V8A at 22 °C); (**G**) Conidiophores (on V8A at 22 °C); (**H**) Conidia (on V8A at 22 °C); (**I**) Conidia (on PCA at 22 °C). Bars: (**E**–**J**) = 25 μm.

**Figure 5 jof-08-00607-f005:**
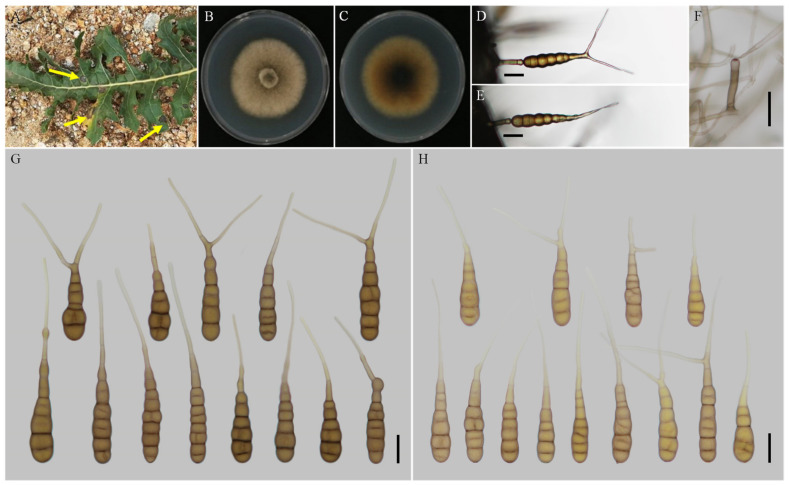
Morphology of *Alternaria neimengguensis* sp. nov. (**A**) Natural symptoms of *Lactuca seriola*; (**B**,**C**) Colony phenotypes (on PDA for 7 days at 25 °C); (**D**,**E**) Sporulation patterns (on V8A at 22 °C); (**F**) Conidiophores (on V8A at 22 °C); (**G**) Conidia (on V8A at 22 °C); (**H**) Conidia (on PCA at 22 °C). Bars: (**D**–**H**) = 25 μm.

**Figure 6 jof-08-00607-f006:**
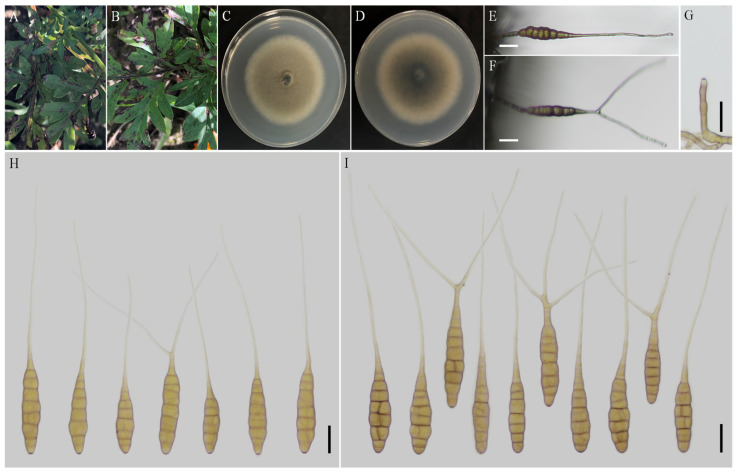
Morphology of *Alternaria sulphureus* sp. nov. (**A**,**B**) Natural symptoms of *Cosmos sulphureus*; (**C**,**D**) Colony phenotypes (on PDA for 7 days at 25 °C); (**E**,**F**) Sporulation patterns (on PCA at 22 °C); (**G**) Conidiophores (on PCA at 22 °C); (**H**) Conidia (on V8A at 22 °C); (**I**) Conidia (on PCA at 22 °C). Bars: (**E**–**I**) = 25 μm.

**Table 1 jof-08-00607-t001:** GenBank accession numbers of *Alternaria* species used in phylogenetic analyses.

Section	Species	Strain	Locality	Substrate	ITS	GAPDH	Alt a 1	TEF1	RPB2
*Porri*	*A. acalyphicola*	CBS 541.94 T	Seychelles	*Acalypha indica*	KJ718097	KJ717952	KJ718617	KJ718446	KJ718271
*Porri*	*A. agerati*	CBS 117221 R	USA	*Ageratum houstonianum*	KJ718098	KJ717953	KJ718618	KJ718447	KJ718272
*Porri*	*A. agripestis*	CBS 577.94 T	Canada	*Euphorbia esula*, stem lesion	KJ718099	JQ646356	KJ718619	KJ718448	KJ718273
*Porri*	*A. allii*	CBS 116701 R	USA	*Allium cepa* var. *viviparum*	KJ718103	KJ717957	KJ718623	KJ718452	KJ718277
*Porri*	*A. alternariacida*	CBS 105.51 T	UK	*Solanum lycopersicum*, fruit	KJ718105	KJ717959	KJ718625	KJ718454	KJ718279
*Porri*	*A. anagallidis*	CBS 117129 R	New Zealand	*Anagallis arvensis*, leaf spot	KJ718109	KJ717962	KJ718629	KJ718457	KJ718283
*Porri*	***A. anhuiensis* sp. nov.**	**YZU 171206 T**	**China**	***Coreopsis basalis*, leaf**	**MK264916**	**MK303949**	**MK303953**	**MK303958**	**MK303960**
*Porri*	*A. anodae*	PPRI 12376	South Africa	*Anoda cristata*, leaf	KJ718110	KJ717963	KJ718630	KJ718458	KJ718284
*Porri*	*A. aragakii*	CBS 594.93 T	USA	*Passiflora edulis*	KJ718111	KJ717964	KJ718631	KJ718459	KJ718285
*Porri*	*A. argyroxiphii*	CBS 117222 T	USA	*Argyroxiphium* sp.	KJ718112	JQ646350	KJ718632	KJ718460	KJ718286
*Porri*	*A. azadirachtae*	CBS 116444 T	Australia	*Azadirachta indica*, leaf spot	KJ718115	KJ717967	KJ718635	KJ718463	KJ718289
*Porri*	*A. bataticola*	CBS 531.63 T	Japan	*Ipomoea batatas*	KJ718117	JQ646349	JQ646433	KJ718465	KJ718291
*Porri*	*A. blumeae*	CBS 117364 T	Thailand	*Blumea aurita*	KJ718126	AY562405	AY563291	KJ718474	KJ718300
*Porri*	*A. calendulae*	CBS 224.76 T	Germany	*Calendula officinalis*	KJ718127	KJ717977	KJ718648	KJ718475	KJ718301
*Porri*	*A. calendulae*	CBS 101498	New Zealand	*Calendula officinalis*, leaf	KJ718128	KJ717978	KJ718645	KJ718476	KJ718302
*Porri*		CBS 116439 T	New Zealand	*Rosa* sp., leaf spot	KJ718129	KJ717979	KJ718646	KJ718477	KJ718303
*Porri*		CBS 116650 R	Japan	*Calendula officinalis*, leaf spot	KJ718130	KJ717980	KJ718647	KJ718478	KJ718304
*Porri*	*A. carthami*	CBS 117091 R	USA	*Carthamus tinctorius*, leaf spot	KJ718133	KJ717983	KJ718651	KJ718481	KJ718307
*Porri*	*A. carthamicola*	CBS 117092 T	Iraq	*Carthamus tinctorius*	KJ718134	KJ717984	KJ718652	KJ718482	KJ718308
*Porri*	*A. cassiae*	CBS 116119 T	Malaysia	*Sauropus androgynus*	KJ718136	KJ717986	KJ718654	KJ718484	KJ718310
*Porri*	*A. catananches*	CBS 137456 T	Netherlands	*Catananche caerulea*	KJ718139	KJ717989	KJ718657	KJ718487	KJ718313
*Porri*	*A. centaureae*	CBS 116446 T	USA	*Centaurea solstitialis*, leaf spot	KJ718140	KJ717990	KJ718658	KJ718488	KJ718314
*Porri*	*A. cichorii*	CBS 102.33 T	Cyprus	*Cichorium intybus*, leaf spot	KJ718141	KJ717991	KJ718659	KJ718489	KJ718315
*Porri*	*A. cirsinoxia*	CBS 113261 T	Canada	*Cirsium arvense*, stem lesion	KJ718143	KJ717993	KJ718661	KJ718491	KJ718317
*Porri*	*A. citrullicola*	CBS 103.32 T	Cyprus	*Citrullus vulgaris*, fruit	KJ718144	KJ717994	KJ718662	KJ718492	KJ718318
*Porri*	***A. coreopsidis* sp. nov.**	**YZU 161159**	**China**	***Coreopsis basalis*, leaf**	**MK264914**	**MK303947**	**MK303951**	**MK303955**	**MK303971**
*Porri*		**YZU 161160 T**	**China**	***Coreopsis basalis*, leaf**	**ON130144**	**ON229924**	**ON229926**	**ON229928**	**ON229930**
*Porri*	*A. crassa*	CBS 110.38 T	Cyprus	*Datura stramonium*, leaf spot	KJ718147	KJ717997	KJ718665	KJ718495	KJ718320
*Porri*		CBS 122590 R	USA	*Datura stramonium*, leaf spot	KJ718152	GQ180072	GQ180088	KJ718500	KJ718325
*Porri*	*A. cucumerina*	CBS 116114 T	USA	*Luffa acutangula*	KJ718153	KJ718000	KJ718668	KJ718501	KJ718326
*Porri*		CBS 117225 R	USA	*Cucumis melo*, leaf spot	KJ718154	KJ718001	KJ718669	KJ718502	KJ718327
*Porri*	*A. cyamopsidis*	CBS 117219 R	USA	*Cyamopsis tetragonoloba*, leaf spot	KJ718157	KJ718004	KJ718672	KJ718505	KJ718330
*Porri*	*A. dauci*	CBS 111.38 T	Italy	*Daucus carota*, seed	KJ718158	KJ718005	KJ718673	KJ718506	KJ718331
*Porri*	*A. deserticola*	CBS 110799 T	Namibia	desert soil	KJ718249	KJ718077	KJ718755	KJ718595	KJ718424
*Porri*	*A. dichondrae*	CBS 199.74 T	Italy	*Dichondra repens*, leaf spot	KJ718166	JQ646357	JQ646441	KJ718514	KJ718339
*Porri*	*A. echinaceae*	CBS 116117 T	New Zealand	*Echinacea* sp., leaf lesion	KJ718170	KJ718015	KJ718684	KJ718518	KJ718343
*Porri*	*A. grandis*	CBS 116695 R	USA	*Solanum tuberosum*, leaf spot	KJ718241	KJ718070	KJ718748	KJ718587	KJ718416
*Porri*	*A. ipomoeae*	CBS 219.79 T	Ethiopia	*Ipomoea batatas*, stem and petiole	KJ718175	KJ718020	KJ718689	KJ718523	KJ718348
*Porri*	*A. jesenskae*	CBS 133855 T	Slovakia	*Fumana procumbens*, seed	KJ718177	KJ718022	KJ718691	KJ718525	KJ718350
*Porri*	*A. linariae*	CBS 105.41 T	Denmark	*Linaria maroccana*, seedling	KJ718180	KJ718024	KJ718692	KJ718528	KJ718353
*Porri*	*A. passiflorae*	CBS 630.93 T	USA	*Passiflora edulis*	KJ718210	JQ646352	KJ718718	KJ718556	KJ718383
*Porri*		CBS 116333 T	New Zealand	*Gaura lindheimeri*, leaf	KJ718211	KJ718046	KJ718719	KJ718557	KJ718384
*Porri*	*A. pipionipisi*	CBS 116115 T	India	*Cajanus cajan*, seed	KJ718214	KJ718049	KJ718722	KJ718560	KJ718387
*Porri*	*A. porri*	CBS 116699 T	USA	*Allium cepa*, leaf spot	KJ718218	KJ718053	KJ718727	KJ718564	KJ718391
*Porri*	*A. protenta*	CBS 116437 T	New Zealand	*Hordeum vulgare*, seed	KJ718220	KJ718055	KJ718729	KJ718566	KJ718393
*Porri*	*A. pseudorostrata*	CBS 119411 T	USA	*Euphorbia pulcherrima*	JN383483	AY562406	AY563295	KC584680	KC584422
*Porri*	*A. ranunculi*	CBS 116330 T	Israel	*Ranunculus asiaticus*, seed	KJ718225	KJ718058	KJ718732	KJ718571	KJ718398
*Porri*	*A. ricini*	CBS 215.31 T	Japan	*Ricinus communis*	KJ718226	KJ718059	KJ718733	KJ718572	KJ718399
*Porri*	*A. rostellata*	CBS 117366 T	USA	*Euphorbia pulcherrima*, leaf	KJ718229	JQ646332	KJ718736	KJ718575	KJ718402
*Porri*	*A. scorzonerae*	CBS 478.83 T	Netherlands	*Scorzonera hispanica*, leaf spot	KJ718191	JQ646334	KJ718699	KJ718538	KJ718364
*Porri*	*A. sennae*	CBS 477.81 T	India	*Senna corymbosa*, leaf	KJ718230	JQ646344	JQ646428	EU130543	KJ718403
*Porri*	*A. sesami*	CBS 115264 R	India	*Sesamum indicum*, seedling	JF780939	KJ718061	KJ718738	KJ718577	KJ718405
*Porri*	*A. sidae*	CBS 117730 T	Kiribati	*Sida fallax*, leaf spot	KJ718232	KJ718062	KJ718739	KJ718578	KJ718406
*Porri*	*A. silybi*	CBS 134092 T	Russia	*Silybum marianum*, leaf	KJ718233	KJ718063	KJ718740	KJ718579	KJ718407
*Porri*		CBS 134093	Russia	*Silybum marianum*, leaf	KJ718234	KJ718064	KJ718741	KJ718580	KJ718408
*Porri*		CBS 134094	Russia	*Silybum marianum*, leaf	KJ718235	KJ718065	KJ718742	KJ718581	KJ718409
*Porri*	*A. solani*	CBS 109157 R	USA	*Solanum tuberosum*, leaf spot	KJ718238	GQ180080	KJ718746	KJ718585	KJ718413
*Porri*	*A. solani-nigri*	CBS 117101 R	New Zealand	*Solanum nigrum*, leaf spot	KJ718247	KJ718075	KJ718753	KJ718593	KJ718422
*Porri*	*A. steviae*	CBS 117362 T	Japan	*Stevia rebaudiana*, leaf spot	KJ718252	KJ718079	KJ718758	KJ718598	KJ718427
*Porri*	*A. tagetica*	CBS 117217 R	USA	*Tagetes* sp., leaf spot	KJ718256	KJ718083	KJ718763	KJ718602	KJ718431
*Porri*		CBS 297.79	UK	*Tagetes* sp., seed	KJ718253	KJ718080	KJ718759	KJ718599	KJ718428
*Porri*		CBS 298.79	UK	*Tagetes* sp., seed	KJ718254	KJ718081	KJ718760	KJ718600	KJ718429
*Porri*		CBS 479.81 R	UK	*Tagetes erecta*, seed	KC584221	KC584143	KJ718761	KC584692	KC584434
*Porri*		CBS 480.81 R	USA	*Tagetes* sp., seed	KJ718255	KJ718082	KJ718762	KJ718601	KJ718430
*Porri*	*A. thunbergiae*	CBS 116331 T	Australia	*Thunbergia alata*, leaf spot	KJ718257	KJ718084	KJ718764	KJ718603	KJ718432
*Porri*	*A. tillandsiae*	CBS 116116 T	New Zealand	*Tillandsia usneoides*	KJ718260	KJ718087	KJ718767	KJ718606	KJ718435
*Porri*	*A. tropica*	CBS 631.93 T	USA	*Passiflora edulis*, fruit	KJ718261	KJ718088	KJ718768	KJ718607	KJ718436
*Porri*	*A. venezuelensis*	CBS 116121 T	Venezuela	*Phaseolus vulgaris*, leaf spot	KJ718263	KJ718263	KJ718770	KJ718609	KJ718438
*Porri*	*A. zinniae*	CBS 117223 R	New Zealand	*Zinnia elegans*, leaf spot	KJ718270	KJ718096	KJ718777	KJ718616	KJ718445
*Porri*		CBS 118.44	Hungary	*Callistephus chinensis*, seed	KJ718264	JQ646361	KJ718771	KJ718610	KJ718439
*Porri*		CBS 117.59	Italy	*Zinnia elegans*	KJ718266	KJ718092	KJ718773	KJ718612	KJ718441
*Porri*		CBS 299.79	UK	*Zinnia* sp., seed	KJ718268	KJ718094	KJ718775	KJ718614	KJ718443
*Gypsophilae*	*A. gypsophilae*	CBS 107.41 T	Netherlands	*Gypsophila elegans*, seed	KC584199	KC584118	KJ718688	KC584660	KC584401

Note: The bold indicate the newly generated sequences. T, ex-type strain; R, representative strain.

**Table 2 jof-08-00607-t002:** Morphological comparisons of the five new *Alternaria* species and their closely related species.

Species	Strain	Conidia				Sporulation Pattern	Medium	Reference
		Shape	Size (μm)	Transversesepta	Beak (μm)			
*A. alternariacida*	CBS 105.51	Smooth-walled, narrowly ovoid; smooth-walled	(85–) 99–111 (–121) × (6–) 7–8 (–10)	(3–) 5–6 (–8)	(47–) 129–257 (–610) × 2	Solitary or in unbranched chains of 2 (–3) conidia	SNA	[4]
***A. anhuiensis* sp. nov.**	**YZU 171206**	**Long-narrow ovoid or ellipsoid; smooth-walled**	**61–100 (–111.5) × (11.5–) 13–19.5**	**6–11**	**(22–) 58–133 (–150.5) × 2.5–4 (–4.5)**	**Solitary**	**V8A**	**This study**
*A. catananches*	CBS 137456	Narrowly ovoid; ornamented in lower half of the conidium	(26–) 37–43 (–57) × (7–) 8–9 (–11)	(2–) 4 (–6)	(77–) 126–160 (–260) × 2	Solitary	SNA	[4]
*A. centaureae*	CBS 116446	Long narrow-ellipsoid or long-ovoid; ornamentation and punctate to pustulate	75–100 × 15–24	7–10	140–190 × 1.5–6	Solitary	V8A	[2]
*A. cichorii*	CBS 117218	Narrow-ovoid or narrow-ellipsoid; smooth-walled	60–80 × 14–18	7–12	120–240 × 2.5–7	Terminal clumps of 4–5 conidia	V8A	[2]
*A. cirsinoxia*	CBS 113261	Long-obclavate, short-ovoid; punctulate-walled	70–90 × 12–22	7–9	80–165 × 2.5–4	Solitary or tufts of 2–7 conidia	V8A	[2]
***A. coreopsidis* sp. nov.**	**YZU 161160**	**Long-narrow ovoid or ellipsoid; smooth-walled**	**(48.5–) 55–80 (–85) × (9–) 10–15**	**6–9**	**(20–) 30–140 (–206) × (2–) 2.5–4**	**Solitary or 2–conidium chains**	**V8A**	**This study**
***A. nanningensis* sp. nov.**	**YZU 171523**	**Ovoid or ellipsoid; smooth-walled**	**(40.5–) 47–79 (–87) × 9–13.5 (–15)**	**6–10 (–11)**	**10–30 × (1–) 1.5–2 (–3)**	**Solitary**	**V8A**	**This study**
***A. neimengguensis* sp. nov.**	**YZU 171784**	**Ovoid or ellipsoid; smooth-walled**	**(70–) 77–130 (–143.5) × (13–) 15–20 (–23)**	**6–11 (–12)**	**(24.5–) 35–65 (–92) × (1.5–) 2–3 (–4)**	**Solitary**	**V8A**	**This study**
*A. obtecta*	CBS 134278	Long-ovoid or ellipsoid; smooth or punctulate-walled	65–95 × 18–22	7–10	55–150 × 2	Solitary	PCA	[2]
*A. porri*	CBS 116698	Ovoid, sometimes broad or nearly cylindrical; smooth or punctulate-walled	70–105 × 19–24	8–12	95–160 × 2–6.5	Solitary	V8A	[2]
*A. silybi*	CBS 134093	Long-ellipsoid, subcylindrical or long-ovoid	50–80 × 15–20 (–22)	(5–) 7–10	70–130 (–190) × 3	Solitary	V4A	[40]
*A. steviae*	CBS 117362	Long-ovoid, subellipsoid, or obovoid; smooth or punctulate-walled	55–95 × 18–30	7–10	60–120 × 1.5–2.5	Solitary or tiny distal clumps	V8A	[2]
***A. sulphureus* sp. nov.**	**YZU 191448**	**Ovoid, ellipsoid, or obovoid; smooth-walled**	**(64–) 74–116 × (12.5–) 14–20 (–25.5)**	**(5–) 7–11**	**(25.5–) 34–151 (–159.5) × 2.5–4.5 (–5.5)**	**Solitary**	**V8A**	**This study**
*A. tillandsiae*	CBS 116116	Long-ovoid, ellipsoid, long-obovoid; smooth or a minor punctulate-walled	70–102 × 16–19	8–11	75–120 × 2	Solitary	V8A	[2]

## Data Availability

The sequences newly generated in this study have been submitted to the GenBank database.

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
