# Peer review of "New Species of Large-Spored Alternaria in Section Porri Associated with Compositae Plants in China"

_jof, 2022, doi:10.3390/jof8060607_

Round 1
Reviewer 1 Report
1: Abstract is very superficial, I prefer to amend it with all relevant information necessary to be in the abstract.
2. Discussion section requires better English and scientific language Language phrasing.
3. Conclusion and future prospects are not provided.
4. Justify the need of the work in the introduction and add a suitable novelty statement.
Author Response
1. Abstract is very superficial, I prefer to amend it with all relevant information necessary to be in the abstract.
Response: It was revised in the manuscript. Please check the Page 1 Line 12-28.
- Discussion section requires better English and scientific language Language phrasing.
Response: The Discussion was carefully checked again and revised. Please re-review it (Page 17-18).
- Conclusion and future prospects are not provided.
Response: It was provided (Page 18 Line 389-399).
- Justify the need of the work in the introduction and add a suitable novelty statement.
Response: It was added in the Introduction. Please check the Page 2 Line 62-66.

Reviewer 2 Report
The paper by Zhao et al. describes the identification of five news Alternaria large-spored species belonging to the section Porri and additional isolates from 3 other sections and one monotypic lineage. All these strains were collected from symptomatic leaves of composite plants in China. Descriptions of these fungal species were performed using both morphological and molecular analyses and in agreement with the taxonomic standards for the genus Alternaria.
No major revision is required, however I have minor comments that could improve the manuscript.
1) It would be interesting to have more information concerning sample collection and fungal isolation: how many plants were sampled? How many fungal isolates were observed? What was the frequency of isolation of the described fungi, in particular the new species? I also regret that 4 of the 5 newly proposed species are described based on aa single specimen.
2) The first part of the discussion (lanes 306 – 311) should be moved to the result section.
3) Fig S1 shows a GPD-RPB tree while Table S1 depicts GPD, ITS and AltA sequences.
4) some sentences are difficult to understand or incomplete:
-lane 25: which is characterized (instead of which are characterized)
-lane 30: Since the 20 th century instead of 20 st
-lanes 31 to 36, this sentence lacks a verb;
-lanes 43-45
-lane 62: what means “of the 19 represent strains”?
-lane 166: which differed instead of which different?
-lane 337-338
Author Response
It would be interesting to have more information concerning sample collection and fungal isolation: how many plants were sampled? How many fungal isolates were observed? What was the frequency of isolation of the described fungi, in particular the new species? I also regret that 4 of the 5 newly proposed species are described based on aa single specimen.
Response: â– Symptomatic samples have been collected all over the country and 14 host plants are related to the present studies (Please check Page 2 Line 69). â– Over ten similar spores were randomly picked from a sample for sub-culturing to obtain the pure cultures, and 2 to 3 strains were selected for deposition when exhibiting similar cultural morphology on PDA. A total of 81 strains were kept in test-tube slants and deposited at 4 °C (Please check Page 2 Line 75-78). â– The new species are only found on the present host. Our continuing study will be lasted in the future and we hope to find more related hosts. â– Indeed, there are more than three strains of each the other four species kept in our fungal herbarium. For each those four species, the five gene sequences and morphology are identical among all the strains. There are some nucleotides differences found among the two strains of A. coreopsidis. Hence two strains were used in this study.
2) The first part of the discussion (lanes 306 – 311) should be moved to the result section.
Response: It was moved. Please check the Page 3 Line 124-132.
3) Fig S1 shows a GPD-RPB tree while Table S1 depicts GPD, ITS and Alt A sequences.
Response: The GAPDH and RPB2 are informatics for the large spore identification. So, the known species were only determined for by the two genes and to know the sections. The ITS sequence information is deleted in the Table S1. Please check it.
4) some sentences are difficult to understand or incomplete:
-lane 25: which is characterized (instead of which are characterized)
Response: It was revised in the manuscript. All the manuscript was carefully checked again and the similar errors are revised.
-lane 30: Since the 20 th century instead of 20 st
Response: It was revised in the manuscript (Line 41).
-lanes 31 to 36, this sentence lacks a verb;
Response: It was revised in the manuscript (Line 42-43).
-lanes 43-45
Response: It was revised in the manuscript.
-lane 62: what means “of the 19 represent strains”?
Response: It is deleted in the manuscript. Actually, that means the total strain number used in the study.
-lane 166: which differed instead of which different?
Response: It was revised in the manuscript (Line 203).
-lane 337-338
Response: It was revised in the manuscript (Line 378-379).
1) It would be interesting to have more information concerning sample collection and fungal isolation: how many plants were sampled? How many fungal isolates were observed? What was the frequency of isolation of the described fungi, in particular the new species? I also regret that 4 of the 5 newly proposed species are described based on aa single specimen.
Response: â– Symptomatic samples have been collected all over the country and 14 host plants are related to the present studies (Please check Page 2 Line 69). â– Over ten similar spores were randomly picked from a sample for sub-culturing to obtain the pure cultures, and 2 to 3 strains were selected for deposition when exhibiting similar cultural morphology on PDA. A total of 81 strains were kept in test-tube slants and deposited at 4 °C (Please check Page 2 Line 75-78). â– The new species are only found on the present host. Our continuing study will be lasted in the future and we hope to find more related hosts. â– Indeed, there are more than three strains of each the other four species kept in our fungal herbarium. For each those four species, the five gene sequences and morphology are identical among all the strains. There are some nucleotides differences found among the two strains of A. coreopsidis. Hence two strains were used in this study.
2) The first part of the discussion (lanes 306 – 311) should be moved to the result section.
Response: It was moved. Please check the Page 3 Line 124-132.
3) Fig S1 shows a GPD-RPB tree while Table S1 depicts GPD, ITS and Alt A sequences.
Response: The GAPDH and RPB2 are informatics for the large spore identification. So, the known species were only determined for by the two genes and to know the sections. The ITS sequence information is deleted in the Table S1. Please check it.
4) some sentences are difficult to understand or incomplete:
-lane 25: which is characterized (instead of which are characterized)
Response: It was revised in the manuscript. All the manuscript was carefully checked again and the similar errors are revised.
-lane 30: Since the 20 th century instead of 20 st
Response: It was revised in the manuscript (Line 41).
-lanes 31 to 36, this sentence lacks a verb;
Response: It was revised in the manuscript (Line 42-43).
-lanes 43-45
Response: It was revised in the manuscript.
-lane 62: what means “of the 19 represent strains”?
Response: It is deleted in the manuscript. Actually, that means the total strain number used in the study.
-lane 166: which differed instead of which different?
Response: It was revised in the manuscript (Line 203).
-lane 337-338
Response: It was revised in the manuscript (Line 378-379).
